# LLM-Rec: Personalized Recommendation via Prompting Large Language Model

## Abstract

Text-based recommendation holds a wide range of practical applications due to its versatility, as textual descriptions can represent nearly any type of item. However, directly employing the original item descriptions as input features may not yield optimal recommendation performance. This limitation arises because these descriptions often lack comprehensive information that can be effectively exploited to align with user preferences. Recent advances in large language models (LLMs) have showcased their remarkable ability to harness common-sense knowledge and reasoning. In this study, we investigate diverse prompting strategies aimed at *augmenting the input text* to enhance personalized text-based recommendations. Our novel approach, coined LLM-Rec, encompasses four distinct prompting techniques: (1) basic prompting, (2) recommendation-driven prompting, (3) engagement-guided prompting, and (4) recommendation-driven + engagement-guided prompting. Our empirical experiments show that incorporating the augmented input text generated by the LLMs yields discernible improvements in recommendation performance. Notably, the recommendation-driven and engagement-guided prompting strategies exhibit the capability to tap into the language model's comprehension of both general and personalized item characteristics. This underscores the significance of leveraging a spectrum of prompts and input augmentation techniques to enhance the recommendation prowess of LLMs.

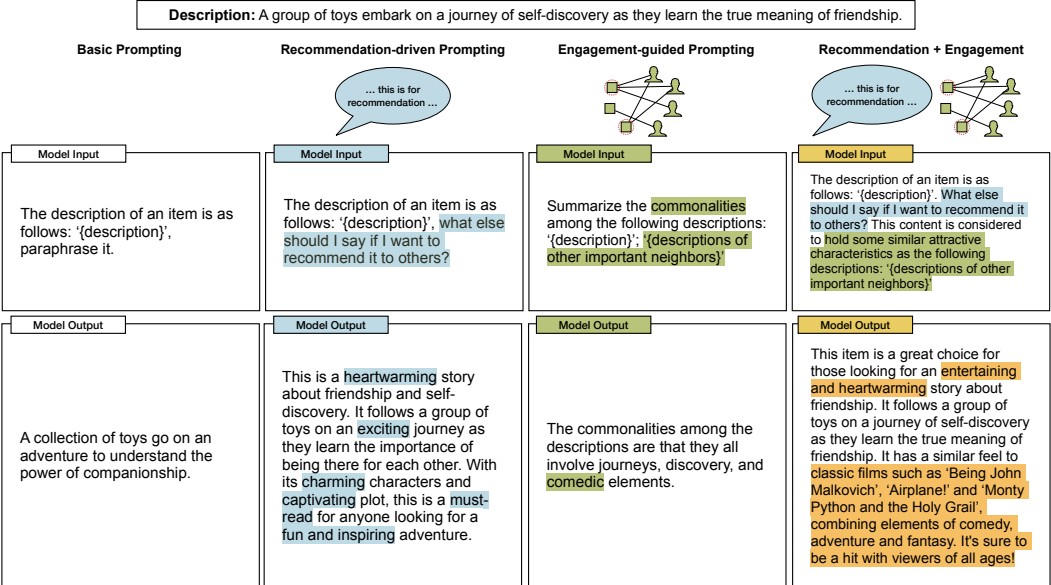

Figure 1: LLM-Rec plays a crucial role in enabling large language models to provide relevant context and help better align with user preferences. Prompts and augmented texts are highlighted.

## 1 INTRODUCTION

Text-based recommendation systems exhibit a broad spectrum of applications, spanning across diverse domains and industries. This versatility mainly stems from the capability of natural language to effectively describe nearly *any* type of items, encompassing not only products, movies, and books but also news articles and user-generated content, including short videos and social media posts (Pazzani & Billsus, 2007; Javed et al., 2021; Poirier et al., 2010; Bai et al., 2022; Wu et al., 2020; Oppermann et al., 2020; Chen et al., 2017; Gupta & Varma, 2017; Wang et al., 2018). Nonetheless, there remains scope for recommendation enhancement, as text-based recommendation systems are frequently challenged by the inherent limitation of **incomplete or insufficient information within item descriptions**, which hinders the task of accurately *aligning* item characteristics with user preferences (Perez et al., 2007; Dumitru et al., 2011). The incompleteness may arise from two sources: a limited comprehension of the items themselves and an insufficient understanding of the users for whom recommendations are generated.

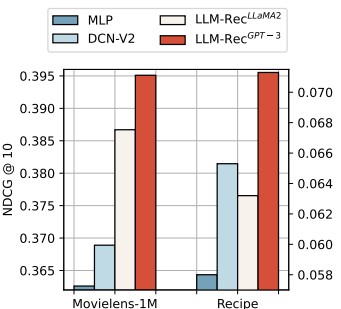

Figure 2: The MLP model integrating the augmented text as its input (*i.e.*, LLM-REC) achieves comparable or even superior recommendation performance compared to both the baseline model (*i.e.*, MLP) and more advanced models like DCN-V2 (Wang et al., 2021) that rely solely on the original item description.

This challenge is not confined only to domains with well-defined and categorized items, such as movies; it also extends to domains characterized by novel, unclassified, or less categorically structured items, as observed in the case of user-generated content. In the context of movie recommendations, a film's description may include the main actors, and a brief plot summary. However, this limited information may not capture crucial elements like genre, tone, cinematography style, or thematic depth. Consequently, a user seeking recommendations for "visually stunning science fiction films" might miss out on relevant selections if the system solely relies on superficial descriptions.

As for user-generated content, imagine a social platform where users regularly post recipes which are often accompanied with brief textual descriptions like the name of the dish and a few ingredients, with limited details regarding preparation time, dietary restrictions, or flavor profiles. Now, consider a user who follows a vegan diet and is interested in discovering new plant-based recipes. Since the user-generated content often lacks comprehensive dietary information and may not explicitly mention terms like "vegan", "plant-based", or "vegetarian", in this scenario, the recommendation system, relying solely on the incomplete descriptions, may struggle to discern the vegan-friendliness of the recipes. Consequently, the user receives recommendations that include non-vegan dishes, ultimately leading to a mismatch between their preferences and the content suggested.

Traditionally, researchers have advocated the augmentation of item descriptions through the incorporation of external knowledge sources (Di Noia et al., 2012; Musto et al., 2018; Sachdeva & McAuley, 2020). Notably, Di Noia et al. (2012) harnesse data from external databases such as `DBpedia` (Bizer et al., 2009), `Freebase` (Bollacker et al., 2008), and `LinkedMDB` (Hassanzadeh & Consens, 2009) to gather comprehensive information pertaining to movies, including details about actors, directors, genres, and categories. This approach aimed to enrich the background knowledge available to movie recommender systems. The explicit semantics embedded in these external knowledge sources have demonstrated a discernible enhancement in recommendation performance (Musto et al., 2017). However, it is essential to acknowledge that this process necessitates a profound domain expertise to effectively and efficiently select and leverage the precise database, ensuring the incorporation of genuinely valuable information into item descriptions (Dumitru et al., 2011).

The recent advances in the development of large language models (LLMs) underscore their exceptional capacity to store comprehensive world knowledge (Peters et al., 2018; Goldberg, 2019; Tenney et al., 2019; Petroni et al., 2019), engage in complex reasoning (Wei et al., 2022; Zhou et al., 2022), and function as versatile task solvers (Zhao et al., 2023; Ouyang et al., 2022; Kaplan et al., 2020). In light of this advancement and recognizing the challenge posed by incomplete item descriptions, our study introduces the LLM-REC framework. This approach is designed to *enrich input text* with the intrinsic capabilities of LLMs for personalized recommendations. By leveraging LLMs, which

have been fine-tuned on extensive language datasets (Ouyang et al., 2022; Touvron et al., 2023a), our goal is to unlock their potential in generating input text that is not only contextually aware but also of high quality (as exemplified in Figure 1), thereby elevating the overall recommendation quality. Through comprehensive empirical experiments, we evaluate the effectiveness of the LLM-REC framework. Figure 2 shows one of these results where LLM-REC enhances the performance of basic MLP (Multi-Layer Perceptron) models, enabling them to attain comparable or even superior recommendation results, surpassing more complex feature-based recommendation approaches. Our study provides insights into the impact of different prompting strategies on recommendation performance and sheds light on the potential of leveraging LLMs for personalized recommendation.

## 2 LLM-REC

Consider two tasks: (1) creating a paragraph that provides a general movie summary and (2) creating a paragraph that provides a movie summary but is specifically intended for generating recommendations. When composing a summary for recommendation purposes, it is customary to infuse it with specific emphases grounded in the author's *comprehension* of the movie. This might involve accentuating the movie's distinctive attributes that set it apart from other movies. For instance, one may opt to incorporate genre information as a crucial element for classifying the movie. However, the decision to leverage the concept of genre for enhancing the summary is predicated on the author's understanding that the genre is a meaningful construct, effectively aligning the summary with the preferences and expectations of the intended audience. This paper aims to explore the potential of large language models when prompted to generate informative item descriptions and subsequently how to leverage this augmented text for enhancing personalized recommendations. Specifically, our study focuses on investigating *four* distinct prompting strategies, namely basic prompting, recommendation-driven prompting, engagement-guided prompting, and the combination of recommendation-driven and engagement-guided prompting.

**Basic Prompting.** The concept of basic prompting closely resembles the task of crafting a general movie summary. Within this scope, we consider three basic prompting variants and refer to them as $p_{para}$, $p_{tag}$, and $p_{infer}$, respectively in the following experiments. $p_{para}$ instructs LLMs to paraphrase the original content description, emphasizing the objective of maintaining the same information without introducing any additional details. Given the original content description, the prompt we use is *"The description of an item is as follows '{description}', paraphrase it."* $p_{tag}$ aims to guide LLMs to summarize the content description by using tags, striving to generate a more concise overview that captures key information. The corresponding prompt is *"The description of an item is as follows '{description}', summarize it with tags."* $p_{infer}$ instructs LLMs to deduce the characteristics of the original content description and provide a categorical response that operates at a broader, less detailed level of granularity. We use the following prompt in the experiments: *"The description of an item is as follows '{description}', what kind of emotions can it evoke?"*

**Recommendation-driven Prompting.** This prompting strategy is to add a recommendation-driven instruction, into the basic prompting, resembling the task of creating a paragraph intended for making recommendations. We refer to the three recommendation-driven prompting as $p_{para}^{rec}$, $p_{tag}^{rec}$, and $p_{infer}^{rec}$, respectively in the following experiments, aligning with their counterparts in the basic prompting strategy. $p_{para}^{rec}$ represents the prompt: *"The description of an item is as follows '{description}', what else should I say if I want to recommend it to others?"* The prompt for $p_{tag}^{rec}$ is *"The description of an item is as follows '{description}', what tags should I use if I want to recommend it to others?"* The prompt for $p_{infer}^{rec}$ is *"The description of an item is as follows '{description}', recommend it to others with a focus on the emotions it can evoke."*

**Engagement-guided Prompting.** As previously elucidated, the deficiency in item descriptions can also emanate from a limited comprehension of the user cohort for whom the recommendations are being generated. Typically, item descriptions are initially formulated for broad, general purposes, devoid of specific targeting toward particular user groups. As a result, they often fall short in capturing the intricate nuances of items required for a more fine-grained alignment with individual user preferences. The goal of the engagement-guided prompting strategy is to leverage user behavior, specifically the interactions between users and items (*i.e.*, user-item engagement) to devise prompts with the intention to steer LLMs towards a more precise comprehension of the attributes within the items, thus generating more insightful and contextually relevant descriptions that align

more closely with the preferences of intended users. We refer to this variant as $p^{eng}$. To create the engagement-guided prompt, we combine the description of the target item, denoted as $d_{target}$, with the descriptions of $T$ **important** neighbor items, represented as $d_1, d_2, \cdots, d_T$. The importance is measured based on user engagement. More details can be found in Appendix A.1.3. The exact prompt of this prompting strategy is *"Summarize the commonalities among the following descriptions: 'description'; 'descriptions of other important neighbors'"*

**Recommendation-driven + Engagement-guided Prompting.** This type of prompt intends to incorporate both the recommendation-driven and engagement-guided instructions, which we denote as $p^{rec+eng}$: *"The description of an item is as follows: 'description'. What else should I say if I want to recommend it to others? This content is considered to hold some similar attractive characteristics as the following descriptions: 'descriptions of important neighbors'"*

**How does** LLM-REC **affect personalized recommendation?** In our experiments, we discover that first and foremost, LLM-REC stands out as a versatile yet simple framework, largely unrestricted by the type of items. Our experimental results on two datasets including the items that are categorically structured and extensively studied to items that are relatively novel and unclassified such as user-generated content, consistently demonstrate the substantial improvement in personalized recommendations. More importantly, this method of input augmentation requires considerably less domain expertise compared to prior studies, making it much more accessible for implementation.

Second, although the efficacy of different LLM-REC prompting components may vary across datasets due to factors such as item characteristics and the quality of original item descriptions, we find that concatenating the text augmented by LLM-REC *consistently* leads to enhanced performance. Simple models, such as MLP, can achieve performance on par with, or even better than, more advanced and complex models. This finding underscores the potential of simplified training to address challenges due to more complex models. In addition, it outperforms other knowledge-based text augmentation methods in the domains that are either well classified or more novel and dynamic.

Third, LLM-REC contributes to increased recommendation transparency and explainability. The ability to directly investigate the augmented text not only enhances our understanding of the recommendation models but also offers insights into the characteristics of the items. It is invaluable for both users and system designers seeking to comprehend the rationale behind recommendations.

## 3 EXPERIMENTS

### 3.1 EXPERIMENT SETUP

Figure 3 shows our architecture. We evaluate the influence of LLM-REC prompting on input augmentation by comparing the recommendation module that integrates the augmented text as input with the same model that only relies on the original content descriptions. Additional details including model training, hyper-parameter settings and implementation details are shown in Appendix A.1.

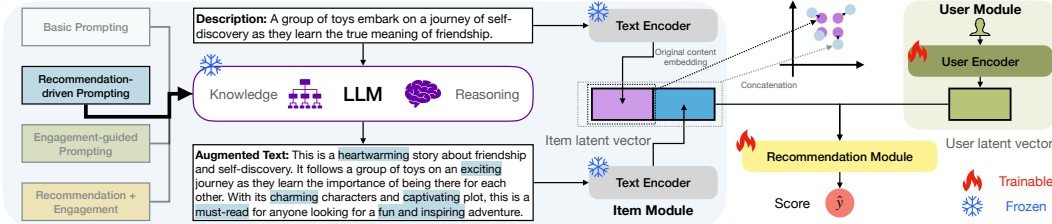

Figure 3: Evaluation architecture. Only prompts and corresponding augmented text are different. Other input and modules *remain the same* throughout the evaluation process.

**Datasets.** Two recommendation benchmarks are used: (1) Movielens-1M (Harper & Konstan, 2015) for movie recommendation, and (2) Recipe (Majumder et al., 2019) for recipe recommendation. The selection of these benchmarks is mainly motivated by two factors. First, Recipes (Majumder et al., 2019) represent user-generated content from social media platforms, resulting in a notably more diverse and less categorically structured item pool compared to movies. Second, while movie descriptions predominantly consist of narrative summaries, recipe descriptions are instructional in nature. The evaluation of LLM-REC on these diverse datasets enables us to gain a comprehensive under-

Table 1: Average recommendation performance among different prompting strategies across five different splits. The best performance among the three Basic Prompting and three Recommendation-driven Prompting strategies is reported. The overall best results are highlighted in **bold**. (Rec: Recommendation-driven; Eng: Engagement-guided; See Appendix A.2.2 for complete results)

| | | **Movielens-1M** | | | **Recipe** | | |
|---|---|---|---|---|---|---|---|
| | | Precision@10 | Recall@10 | NDCG@10 | Precision@10 | Recall@10 | NDCG@10 |
| Content Description | | $0.2922_{\pm0.0019}$ | $0.2455_{\pm0.0031}$ | $0.3640_{\pm0.0039}$ | $0.0325_{\pm0.0021}$ | $0.0684_{\pm0.0066}$ | $0.0580_{\pm0.0054}$ |
| LLAMA-2 | Basic | $0.3006_{\pm0.0018}$ | $0.2570_{\pm0.0036}$ | $0.3754_{\pm0.0033}$ | $0.0353_{\pm0.0024}$ | $0.0751_{\pm0.0067}$ | $0.0641_{\pm0.0058}$ |
| | Rec | $0.3025_{\pm0.0027}$ | $0.2601_{\pm0.0030}$ | $0.3784_{\pm0.0047}$ | $0.0344_{\pm0.0029}$ | $0.0739_{\pm0.0083}$ | $0.0617_{\pm0.0063}$ |
| | Eng | $0.2989_{\pm0.0019}$ | $0.2546_{\pm0.0039}$ | $0.3736_{\pm0.0033}$ | $0.0333_{\pm0.0027}$ | $0.0709_{\pm0.0077}$ | $0.0600_{\pm0.0057}$ |
| | Rec+Eng | $0.2977_{\pm0.0010}$ | $0.2525_{\pm0.0021}$ | $0.3720_{\pm0.0022}$ | $0.0334_{\pm0.0025}$ | $0.0704_{\pm0.0073}$ | $0.0593_{\pm0.0062}$ |
| GPT-3 | Basic | $0.3001_{\pm0.0027}$ | $0.2569_{\pm0.0028}$ | $0.3747_{\pm0.0042}$ | $0.0356_{\pm0.0024}$ | $0.0754_{\pm0.0089}$ | $0.0644_{\pm0.0068}$ |
| | Rec | $0.3025_{\pm0.0023}$ | $0.2577_{\pm0.0053}$ | $0.3786_{\pm0.0041}$ | $\mathbf{0.0361}_{\pm0.0031}$ | $\mathbf{0.0771}_{\pm0.0086}$ | $\mathbf{0.0649}_{\pm0.0069}$ |
| | Eng | $0.3036_{\pm0.0020}$ | $0.2608_{\pm0.0030}$ | $0.3801_{\pm0.0032}$ | $0.0348_{\pm0.0031}$ | $0.0732_{\pm0.0088}$ | $0.0628_{\pm0.0077}$ |
| | Rec+Eng | $\mathbf{0.3038}_{\pm0.0020}$ | $\mathbf{0.2603}_{\pm0.0042}$ | $\mathbf{0.3802}_{\pm0.0037}$ | $0.0349_{\pm0.0024}$ | $0.0732_{\pm0.0066}$ | $0.0625_{\pm0.0060}$ |

standing of how different prompting strategies influence recommendation outcomes. For additional dataset statistics, sample examples, and preprocessing specifics, please refer to Appendix A.1.1.

**Language Models.** We prompt two large language models to augment item descriptions. The first is GPT-3 (Brown et al., 2020), particularly its variant `text-davinci-003`. This model is an advancement over the InstructGPT models (Ouyang et al., 2022). We select this variant due to its ability to consistently generate high-quality writing, effectively handle complex instructions, and demonstrate enhanced proficiency in generating longer form content (Raf, 2023). The second is LLAMA-2 (Touvron et al., 2023b), which is an open-sourced model that has shown superior performance across various external benchmarks in reasoning, coding, proficiency, and knowledge tests. Specifically, for our experiments, we use the LLAMA-2-CHAT variant of 7B parameters.

**Evaluation Protocols.** We adopt the evaluation methodology of Wei et al. (2019). We randomly divide the dataset into training, validation, and test sets using an 8:1:1 ratio. Negative training samples are created by pairing users and items without any recorded interactions (note that these are pseudo-negative samples). For the validation and test sets, we pair each observed user-item interaction with 1,000 items that the user has not previously interacted with. It is important to note that there is *no* overlap between the negative samples in the training set and the unobserved user-item pairs in the validation and test sets. This ensures the independence of the evaluation data. We use metrics such as Precision@K, Recall@K and NDCG@K to evaluate the performance of top-K recommendations, where $K = 10$. We report the average scores across five different splits of the testing sets. The recommendation module is the combination of an MLP model and a dot product.

## 4   RESULTS

**Incorporating text augmentation through large language models prompted by LLM-REC consistently boosts recommendation performance.** We compare the recommendation performance of the models using the concatenated embeddings of content descriptions and prompt responses as their input against models relying solely on content description embeddings. The results, presented in Table 1, reveal a noteworthy and consistent enhancement in recommendation performance across various prompting strategies within two benchmark datasets. For instance, LLM-REC prompting yields relative gains in NDCG@10 ranging from $2.20\%$ to $4.45\%$ in Movielens-1M and from $2.24\%$ to $11.72\%$ in Recipe. These substantial improvements underscore the effectiveness of LLM-REC in guiding large language models to augment item descriptions.

**LLM-REC empowers simple MLP models to achieve comparable or even superior recommendation performance, surpassing other more complex feature-based recommendation methods.** Table 2 shows the average recommendation performance between LLM-REC and baseline approaches across five different splits. The rows of LLM-REC indicate the results of the MLP models that take the concatenation of all augmented text and original content descriptions as input. Except for Item Popularity, other baselines takes the original content descriptions as their input. We have selected five baseline recommendation models for comparison. The first baseline relies solely on item popularity and does not involve any learning process; we refer to it as Item Popularity. The second baseline combines an MLP with a dot product, and for simplicity, we refer to it as MLP. Furthermore, we choose three more advanced, feature-based recommendation models. AutoInt (Song

Table 2: Average recommendation performance between LLM-REC and baseline approaches across five different splits. The best results are highlighted in **bold**, the second-best results are underlined, and relative gains compared to the MLP baseline are indicated in green.

| | | Movielens-1M | | | Recipe | | |
| | | Precision@10 | Recall@10 | NDCG@10 | Precision@10 | Recall@10 | NDCG@10 |
|---|---|---|---|---|---|---|---|
| Item Popularity | | $0.0426_{\pm0.0019}$ | $0.0428_{\pm0.0028}$ | $0.0530_{\pm0.0035}$ | $0.0116_{\pm0.0025}$ | $0.0274_{\pm0.0083}$ | $0.0201_{\pm0.0053}$ |
| MLP | | $0.2922_{\pm0.0019}$ | $0.2455_{\pm0.0031}$ | $0.3640_{\pm0.0039}$ | $0.0325_{\pm0.0021}$ | $0.0684_{\pm0.0066}$ | $0.0580_{\pm0.0054}$ |
| AutoInt (Song et al., 2019) | | $0.2149_{\pm0.0078}$ | $0.1706_{\pm0.0075}$ | $0.2698_{\pm0.0092}$ | $0.0351_{\pm0.0032}$ | $0.0772_{\pm0.0102}$ | $\underline{0.0658}_{\pm0.0089}$ |
| DCN-V2 (Wang et al., 2021) | | $0.2961_{\pm0.0050}$ | $0.2433_{\pm0.0057}$ | $0.3689_{\pm0.0033}$ | $\underline{0.0360}_{\pm0.0036}$ | $\underline{0.0786}_{\pm0.0104}$ | $0.0653_{\pm0.0085}$ |
| EDCN (Chen et al., 2021) | | $0.2935_{\pm0.0036}$ | $0.2392_{\pm0.0051}$ | $0.3678_{\pm0.0053}$ | $0.0354_{\pm0.0030}$ | $0.0772_{\pm0.0091}$ | $0.0652_{\pm0.0071}$ |
| KAR (Xi et al., 2023) | | $0.3056_{\pm0.0026}$ | $0.2623_{\pm0.0034}$ | $0.3824_{\pm0.0042}$ | $0.0298_{\pm0.0018}$ | $0.0611_{\pm0.0049}$ | $0.0525_{\pm0.0043}$ |
| - augmented with ground truth | | $0.3075_{\pm0.0015}$ | $0.2636_{\pm0.0035}$ | $0.3853_{\pm0.0027}$ | - | - | - |
| LLM-REC | LLAMA-2 | $\underline{0.3102}_{\pm0.0014}$ (+6.16%) | $\underline{0.2712}_{\pm0.0026}$ (+10.47%) | $\underline{0.3867}_{\pm0.0027}$ (+6.24%) | $0.0359_{\pm0.0024}$ (+10.46%) | $0.0770_{\pm0.0076}$ (+12.57%) | $0.0632_{\pm0.0052}$ (+8.97%) |
| | GPT-3 | $\mathbf{0.3150}_{\pm0.0023}$ (+7.80%) | $\mathbf{0.2766}_{\pm0.0030}$ (+12.67%) | $\mathbf{0.3951}_{\pm0.0035}$ (+8.54%) | $\mathbf{0.0394}_{\pm0.0033}$ (+21.23%) | $\mathbf{0.0842}_{\pm0.0098}$ (+23.10%) | $\mathbf{0.0706}_{\pm0.0084}$ (+21.72%) |

et al., 2019) is a multi-head self-attentive neural network with residual connections designed to explicitly model feature interactions within a low-dimensional space. DCN-V2 (Wang et al., 2021) represents an enhanced version of DCN (Wang et al., 2017) and incorporates feature crossing at each layer. Lastly, EDCN (Chen et al., 2021) introduces a bridge module and a regulation module to collaboratively capture layer-wise interactive signals and learn discriminative feature distributions for each hidden layer in parallel networks, such as DCN.

**LLM-REC augmentation outperforms other text augmented methods for recommendation.** We compare LLM-REC with one of the most recent advancements in the field of using LLMs to augment item information, specifically Knowledge Augmented Recommendation (KAR) as proposed by Xi et al. (2023). KAR introduces a fusion of domain knowledge and prompt engineering to generate factual knowledge pertaining to the items (for detailed implementation information, see Appendix A.1.7). In contrast to KAR's approach, LLM-REC places a particular emphasis on the innate common-sense reasoning capabilities of large language models and notably does not mandate domain expertise. Since the augmented information may not necessarily be correct, we further implement a variant with ground truth knowledge. It aligns with strategies akin to those introduced by Di Noia et al. (2012), who harnessed external databases to enhance item information. In a manner consistent with this approach, we incorporate genre information into the item descriptions. It is noteworthy that genre constitutes one of the metadata components in the Movielens-1M dataset. Such categorical characteristics are absent in the Recipe dataset. As a result, we exclusively apply this variant to the Movielens-1M dataset. As shown in Table 2, the incorporation of knowledge-based text augmentation offers significant improvements in recommendation performance for well-classified items, such as movies. However, it becomes evident that this approach faces limitations when applied to items, like user-generated content, that are inherently more novel and dynamic in nature. LLM-REC outperforms them as it emphasizes more on the reasoning ability of LLMs instead of solely considering them as external knowledge sources. More importantly, LLM-REC does not require domain knowledge throughout the entire process.

**What extra information does recommendation-driven strategy prompt LLMs to augment?** We conduct a case study comparing $P_{para}$ with $P_{para}^{rec}$. More specifically, we focus on the items that the recommendation is correct based on the response of $P_{para}^{rec}$ while incorrect based on the response of $P_{para}$. The item descriptions and the corresponding generated responses of the top three such items are shown in Figure 4. Example responses of $P_{tag}$, $P_{tag}^{rec}$, $P_{infer}$, and $P_{infer}^{rec}$ can be found in Appendix A.2. We find that the most distinctive words in the response of $P_{para}^{rec}$ are the words that are related with user preferences. These words include the words that can express users' preferences about items such as exciting, thought-provoking, delicious, and so on. We also discover words that are related to the pre-defined concept in terms of user preferences such as genres (*e.g.*, classic, action, easy-to-make). We hypothesize that the extra words generated with the recommendation-driven prompting strategy improve recommendation performance.

To validate this hypothesis, we design two variants of the response, namely $P_{para}^{mask}$ and $P_{para}^{keyword}$. To construct $P_{para}^{mask}$, we mask the words that appear in the response of $P_{para}^{rec}$ but are absent in the response of $P_{para}$. To construct $P_{para}^{keyword}$, we append the words that (1) appear in the response of $P_{para}^{rec}$ and (2) are pre-defined user-preference-related words such as genres to the end of the response of $P_{para}$ (see Appendix A.1.8 for how we construct these related words). These two variants

**Movielens-1M**

**Description:** A troubled child psychologist helps a young boy who is able to see and communicate with the dead.

$P_{para}$: A child psychologist works with a young boy who has the ability to perceive and talk to spirits.

$P^{rec}_{para}$: This movie is a must-watch for anyone who loves psychological thrillers. It follows the story of a child psychologist as he helps a young boy who can see and communicate with the dead. The movie is full of suspense and mystery, and will keep you on the edge of your seat. It's a great watch for anyone looking for an exciting and thought-provoking movie.

**Description:** Marty McFly must travel back in time to save his future family and ensure his own existence.

$P_{para}$: Marty McFly must go back in time to protect his relatives and guarantee his own survival.

$P^{rec}_{para}$: This item is a must-have for any fan of the classic movie, Back to the Future. It follows Marty McFly as he travels back in time to save his future family and ensure his own existence. With an exciting plot and memorable characters, this item is sure to provide hours of entertainment.

**Description:** Indiana Jones embarks on a thrilling adventure to find the lost Ark of the Covenant and prevent it from falling into the hands of the Nazis.

$P_{para}$: Indiana Jones sets off on an exciting journey to locate the Ark of the Covenant and keep it away from the Nazis.

$P^{rec}_{para}$: This classic action-adventure movie is a must-see for any fan of the Indiana Jones franchise. Follow Indy as he races against time to find the Ark of the Covenant and keep it out of the hands of the Nazis. With its thrilling plot, iconic characters, and stunning visuals, Indiana Jones and the Raiders of the Lost Ark is an unforgettable cinematic experience.

**Recipe**

**Description:** A favorite recipe i found online, and thought i should pass on here at the zarr! this really is a elegant, pretty, but most of all tasty dessert treat.

$P_{para}$: I discovered a delicious recipe online and wanted to share it with everyone at the Zarr. This dessert is both visually appealing and scrumptious.

$P^{rec}_{para}$: I highly recommend this recipe! It's a great combination of elegance, beauty, and deliciousness. The flavors blend together perfectly and it's sure to be a hit with everyone who tries it. Plus, it's easy to make and doesn't require any special ingredients. Give it a try - you won't regret it!

**Description:** i have adopted this recipe from mean chef. this is one of my favorite recipes. the pork is sweet, succulent and falling apart tender. fabulous served alone, on rolls, in tortillas.

$P_{para}$: I took this recipe from a talented cook. It's one of my favorite dishes; the pork is incredibly juicy and tender. It's great to eat on its own, in sandwiches, or in tacos.

$P^{rec}_{para}$: I highly recommend this recipe from the mean chef! The pork is so tender and flavorful, it's sure to be a hit with everyone. Serve it alone, on rolls, or in tortillas for a delicious meal that will have your guests coming back for more.

**Description:** i found this recipe in a taste of home publication awhile back. i had written it out and stuck it in my 'to try' binder. a few months ago i made it for the first time... it's now my absolute favorite baked macaroni and cheese recipe! (with it tweaked for my own tastes). enjoy!

$P_{para}$: This recipe was discovered in a Taste of Home publication and written down to be tried later. After making it for the first time, it quickly became a favorite and was tweaked to suit personal tastes.

$P^{rec}_{para}$: I highly recommend this recipe for baked macaroni and cheese! It's incredibly flavorful and creamy, and it's sure to be a hit with your family and friends. Plus, it's easy to make and customize to your own tastes. Give it a try - you won't regret it!

Figure 4: Example responses generated by GPT-3. The additional information augmented via the recommendation-driven prompting is highlighted in blue. We choose the example responses generated by GPT-3 for illustration. Examples generated by LLAMA-2 can be found in Appendix A.2.3.

of the responses are then fed into MLP models to form baselines. Figure 3 shows the recommendation performances. Comparing the performance of $P^{rec}_{para}$ and $P^{mask}_{para}$, we observe a discernible decline in recommendation performance when words unique to the response of $P^{rec}_{para}$ are selectively masked. This outcome highlights the pivotal role played by the supplementary insights introduced through the augmented text. Furthermore, our investigation reveals that the incorporation of vital keywords, as opposed to the inclusion of all response words, can yield even superior recommendation performance. This phenomenon may be attributed to potential discrepancies or extraneous elements within the response of $P^{rec}_{para}$.

**What extra information does engagement-guided strategy prompt LLMs to augment?** Consistent with our previous experiments, we curate exemplary responses obtained from $p^{eng}$ for closer examination (Figure 5). Our analysis reveals a distinct pattern compared to what we have observed with recommendation-driven prompting. There are primarily two scenarios to consider. First, if the descriptions of the important neighbor items and the target items exhibit high similarity, the impact of $p^{eng}$ resembles that of $p_{para}$, as exemplified in the second Recipe example in Figure 5. Second, $p^{eng}$ guides LLMs to generate additional information, which may be derived from the descriptions of the important neighbor items. Consequently, how the engagement-guided strategy influences LLMs' text generation—whether it aligns with one of the behaviors we have described, both of them, or even other unexplored patterns—largely depends on the composition of the important neighbor items. This composition, in turn, is contingent on the neighbor sampling method which is out of the scope of our study. We leave a more in-depth exploration of this topic to future research endeavors.

Table 3: Average NDCG@10 across five splits.

|  | Movielens-1M | Recipe |
|---|---|---|
| $P_{para}$ | 0.3746 | 0.0611 |
| $P^{keyword}_{para}$ | 0.3822 | 0.0615 |
|  | (+2.03%) | (+0.65%) |
| $P^{rec}_{para}$ | 0.3777 | 0.0646 |
| $P^{mask}_{para}$ | 0.3769 | 0.0611 |
|  | (-0.21%) | (-0.52%) |

Interestingly, the recommendation-driven + engagement-guided prompting strategy is able to generate text that shares similar characteristics with both sub-strategies. How they quantitatively form the final generation remains an open question. Examples can be found in Appendix A.2.3.

**How does concatenating the augmented responses affect recommendation?** In Table 2, we show that the MLP model, which combines all augmented text with the original description embeddings, outperforms more advanced models that rely solely on the original description embeddings as input. Now we take a deeper look at the quality of the combined augmented text. We employ the same recommendation module (*i.e.*, an MLP with a dot product) and evaluate the recommendation performance of various concatenation combinations. The results are illustrated in Figure 6. In Figure 6, the

**Movielens-1M**

**Description:** The Paper Chase follows a first-year Harvard Law student as he struggles to balance his studies and personal life while facing the demanding expectations of his professor.

**Peng:** The commonalities among these descriptions are that they all feature characters facing difficult challenges and obstacles. They also all have themes of resilience, hope, and friendship. We recommend these stories to others because they offer inspiring messages of overcoming adversity and the power of friendship.

**Description:** Under the Rainbow is a comedy about a group of misfit spies who must save a Hollywood hotel from a group of Nazi agents.

**Peng:** The commonalities among these descriptions are that they all involve comedic elements, misfit characters, and a mission to save something from danger. All of these movies are recommended for their humorous take on the genres they represent and their entertaining storylines.

**Description:** A brilliant engineer and his team must race against time to prevent a catastrophic chain reaction from destroying the world.

**Peng:** The commonalities among these descriptions are that they all involve a protagonist who must prevent a catastrophic event from occurring. They also all involve a sense of urgency and suspense. We recommend these stories to others as they provide an exciting and thrilling experience.

**Recipe**

**Description:** I love this crock-pot chicken recipe for two reasons: one, my family loves it and two, it is so easy to make! i got this recipe from my sister. she has two children of her own, and they love it too! it's also the best leftovers in the world -- if there are any!

**Peng:** The commonalities among the descriptions are that they all describe recipes that are easy to make, have great flavor, and are loved by family members. They also all mention leftovers, and two of them mention cutting back on saltiness.

**Description:** easy to assemble, deliciously creamy and a great choice for once-a-month cooking. and there is no canned cream soup in my enchiladas -- hope you give them a try!

**Peng:** The commonalities among the descriptions are that they are all easy to assemble, deliciously creamy, and a great choice for once-a-month cooking. Additionally, none of the recipes involve canned cream soup.

**Description:** this is the perfect chocolate pudding. it has a rich chocolate flavor, great texture and is not high in fat. i can't count the number of times i have made this. i hope you enjoy it as much as we do.

**Peng:** The commonalities among the descriptions are that they all describe recipes that are easy to make, have great flavor and texture, and can be varied with different add-ins

Figure 5: Example responses generated by GPT-3. The additional information augmented via the engagement-guided prompting is colored green. We choose the example responses generated by GPT-3 for illustration. Examples generated by LLAMA-2 can be found in Appendix A.2.3.

model denoted as `Basic` uses the embeddings of text augmented through $p_{para}$. `Concat-Basic` represents the model that concatenates the embeddings of the input text augmented by all Basic Prompting variants. Additionally, `Concat-Rec` is the model that employs the concatenation of the embeddings of input text augmented by all Recommendation-driven Prompting variants. Lastly, `Concat-All` stands for the model that combines the embeddings of input text augmented by all four prompting strategies. Our findings reveal that concatenating more information *consistently* enhances recommendation performance. This emphasizes the added value of incorporating augmented text as opposed to relying solely on the original content description. Additional experiments on other ways of concatenating the augmented text can be found in Appendix A.2.4.

## 5 DISCUSSIONS

In this study, we have investigated the effectiveness of LLM-REC as a simple yet impactful mechanism for improving recommendation through large language models. Our findings reveal several key insights. First, we demonstrate that by combining augmented text with the original description, we observe a significant enhancement in recommendation performance. It also empowers simple models such as MLPs to achieve comparable or even superior recommendation performance than other more complex feature-based methods. Compared with other knowledge-based text augmentation methods, LLM-REC demonstrates superior generalizability. This exceptional performance holds true whether the items under consideration are well-classified or belong to the category of more novel and less-studied items. What distinguishes LLM-REC further is its capacity to operate without the need for domain-specific knowledge throughout the entire process. The emphasis on common-sense reasoning and its domain-agnostic nature makes LLM-REC a versatile and effective choice for recommen-

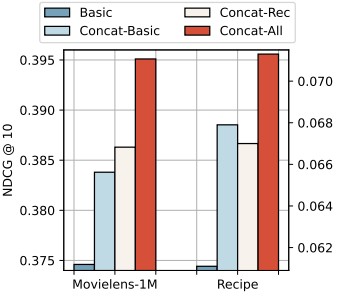

Figure 6: The ablation study shows that recommendation benefits from concatenating the embeddings of the input text augmented by LLM. Full results in Appendix A.2.2.

dation tasks across a broad spectrum of item categories. Furthermore, our experimental results on recommendation-driven and engagement-guided prompting strategies illustrate their ability to encourage the large language model to generate high-quality input text specifically tailored for recommendation purposes. These prompting strategies effectively leverage recommendation goals and user engagement signals to guide the model towards producing more desirable recommendations.

## 6 RELATED WORK

**Augmentation for Text-based Recommendation.** Text-based recommendation systems leverage natural language processing and machine learning techniques to provide personalized recommendations to users based on textual information (Lops et al., 2019; Qiang et al., 2020). However, the performance of such systems can be compromised when dealing with incomplete or insufficient textual information. To address this limitation, several studies have suggested strategies for enhancing textual information. For instance, Li et al. (2010) proposed to extract contextual cues from online reviews, leveraging these narratives to uncover users' preferences and underlying factors influencing their choices (Sachdeva & McAuley, 2020). Other approaches infer linguistic attributes from diverse sources, including emotion, sentiment, and topic, to refine the modeling of both items and users (Sun et al., 2015; Sailunaz & Alhajj, 2019; Ramage et al., 2010; Chen et al., 2010). Furthermore, some works explore the integration of external knowledge bases to enrich the contextual understanding of items (Di Noia et al., 2012; Musto et al., 2018). In a more recent development, Bai et al. (2022) introduced an approach that employs pre-trained language models to generate additional product attributes, such as product names, to augment item contextual information. Diverging from these prior approaches, our contribution is the LLM-REC framework, which employs large language models to enhance input text, providing a versatile solution for personalized recommendations.

**LLM for Recommendation.** The use of large language models in recommender systems has garnered significant attention in recent research. Many studies have explored the direct use of LLMs as recommender models. The underlying principle of these approaches involves constructing prompts that encompass the recommendation task, user profiles, item attributes, and user-item interactions. These task-specific prompts are then presented as input to the LLMs, which is instructed to predict the likelihood of interaction between a given user and item (Dai et al., 2023b; Gao et al., 2023; Geng et al., 2022; Li et al., 2023; Liu et al., 2023b; Zhang et al., 2023). For instance, Wang & Lim (2023) designed a three-step prompting strategy to directly guide LLMs to capture users' preferences, select representative previously interacted items, and recommend a ranked list of 10 items. While these works demonstrate the potential of LLMs as powerful recommender models, the focus primarily revolves around utilizing the LLMs directly for recommendation purposes. However, in this study, we approach the problem from a different perspective. Rather than using LLMs as recommender models, this study explores diverse prompting strategies to *augment input text* with LLMs for personalized content recommendation.

**LLM Augmentation for Recommendation.** Due to LLMs' remarkable text generation ability, many studies have leveraged LLMs as a data augmentation tool (Dai et al., 2023a; Li et al., 2022). Liu et al. (2023a) used an LLM to produce multimodal language-image instruction-following datasets. Through a process of instruction tuning using this generated data, their proposed framework demonstrated an impressive aptitude in advancing vision and language comprehension. There have also been efforts to use LLMs to augment the input side of personalized recommendation. For instance, Chen (2023) incorporated user history behaviors, such as clicks, purchases, and ratings, into LLMs to generate user profiles. These profiles were then combined with the history interaction sequence and candidate items to construct the final recommendation prompt. LLMs were subsequently employed to predict the likelihood of user-item interaction based on this prompt. Xi et al. (2023) introduced a method that leverages the reasoning knowledge of LLMs regarding user preferences and the factual knowledge of LLMs about items. However, our study focuses specifically on using LLMs' knowledge and reasoning ability to generate augmented input text that better captures the characteristics and nuances of items, leading to improved personalized recommendations.

## 7 CONCLUSIONS

We introduced LLM-REC, which enhances personalized recommendation via prompting large language models. We observed from extensive experiments that combining augmented input text and original content descriptions yields notable improvements in recommendation quality. These findings show the potential of using LLMs and strategic prompting techniques to enhance the accuracy and relevance of personalized recommendation with an easier training process. By incorporating additional context through augmented text, we enable the recommendation algorithms to capture more nuanced information and generate recommendations that better align with user preferences.

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
