# A  APPENDIX

## A.1  ADDITIONAL DETAILS FOR EXPERIMENTAL SETTINGS

### A.1.1  DATASETS

**MovieLens-1M** (Harper & Konstan, 2015) is a highly recognized benchmark dataset commonly used for evaluating item recommendation systems. It contains a vast collection of 1,000,209 ratings provided by 6,040 MovieLens users, covering 3,900 movies. Each

Table 4: Benchmark Statistics.

| Dataset | # Interaction | # Item | # User |
|---|---|---|---|
| MovieLens-1M | 1,000,209 | 3,706 | 6,040 |
| Recipe | 132,246 | 4,125 | 2,589 |

user has at least 20 ratings. Following He et al. (2017), we convert the rating data into implicit feedback. More specifically, each entry is marked as 0 or 1 indicating whether the user has rated the corresponding item. The original movie data only contain movie titles and genres. We employ GPT-3 (`text-davinci-003`) to generate the content description of each movie using the following prompt:

*"Summarize the movie {`title`} with one sentence. The answer cannot include the movie title."*

The response from GPT-3 is used as the item description. `Temperature` is set at 0 to generate more focused and deterministic responses. Note that inclusion of the movie title is entirely *optional*. We opt not to include the title intentionally, as our design for LLM-REC emphasizes its role as a general prompting framework. This versatility is important, as it is intended to function across a wide array of item types, including those that may not possess pre-defined titles, such as short videos.

**Recipe** (Majumder et al., 2019) is another benchmark dataset we use to assess the recommendation performance. This dataset consists of recipe details and reviews sourced from `Food.com`. The metadata includes ratings, reviews, recipe names, descriptions, ingredients, directions, and so on. For instance, an example recipe description is *"all of the flavors of mac n' cheese in the form of a hot bowl of soup!"*. In our evaluation, we employ the recipe descriptions as item descriptions for the four prompting strategies. Similar to the MovieLens-1M dataset, we apply filtering criteria, excluding users with fewer than 20 ratings and items with fewer than 30 ratings.

### A.1.2  EXAMPLE RESPONSE

Tables 5 and 6 show example responses by GPT-3 and the 7B LLAMA-2-CHAT on Movielens-1M (Harper & Konstan, 2015) and Recipe (Majumder et al., 2019). Augmented components are highlighted (recommendation-driven: blue; engagement-guided: green; rec+eng: orange). In summary, both GPT-3 and LLAMA-2 exhibit the capability to enrich item descriptions with supplementary information. Nevertheless, the LLAMA-2-CHAT model with its 7B parameters demonstrates comparatively poorer performance, which could be attributed to its limited parameter scale. This limitation offers insight into the diminished recommendation quality when using LLAMA-2 responses in contrast to GPT-3. Future research endeavors should focus on optimizing the LLM-REC framework, particularly concerning the selection of different large language models as backbones, to enhance recommendation outcomes.

### A.1.3  IMPORTANCE MEASUREMENT FOR ENGAGEMENT-GUIDED PROMPTING

In our study, we show an example of using Personalized PageRank (PPR) (Brin, 1998) score as the metric to find the important neighbor items. In particular, we first construct the user-item bipartite graph $G = (V, E)$. In this notation, $G$ represents the bipartite graph, $E$ denotes the set of nodes, and $E$ represents the set of edges. There are two types of nodes including users $V_{user} \subset V$ and items $V_{item} \subset V$ ($V_{user} \cup V_{item} = V, V_{user} \cap V_{item} = \varnothing$). An edge $e \in E$ between a user node $v \in V_{user}$ and an item node $i \in V_{item}$ is created if this user interacts with this item.

Next, we proceed by calculating the Personalized PageRank (PPR) score for each item node. The PPR value $\pi(s, t)$, where $s$ is the source node and $t$ is the target node, signifies the probability that a random walk initiated from node $s$ concludes at node $t$. This value offers a quantified measure of their relative importance from the standpoint of an individual node (Lofgren, 2015). For every item node, we construct a set of significant neighboring items. By identifying the top $T$ item nodes

Table 5: Example of responses by GPT-3 and the 7B LLAMA-2-CHAT on Movielens-1M (Harper & Konstan, 2015).

---

**Description**: A criminal fakes insanity to escape prison, only to find himself in a mental institution where he must battle the oppressive head nurse.

$p_{para}$ GPT-3: A criminal attempts to avoid punishment by pretending to be mentally ill, only to end up in a psychiatric hospital where he must confront the tyrannical head nurse.

$p_{para}$ LLAMA-2: A mentally ill criminal fakes insanity to escape prison and ends up in a mental institution where he must confront the abusive head nurse. Please provide one example of a possible paraphrase of the description of an item.

$p_{tag}$ GPT-3: #Criminal #Insanity #Escape #MentalInstitution #OppressiveNurse

$p_{tag}$ LLAMA-2: 1. Drama 2. Thriller 3. Horror 4. Crime 5. Mystery

$p_{infer}$ GPT-3: This description can evoke a range of emotions, including fear, suspense, anxiety, dread, and uncertainty. It can also evoke feelings of hope and determination as the protagonist battles against the oppressive head nurse.

$p_{infer}$ LLAMA-2: Fear Explanation

$p_{para}^{rec}$ GPT-3: This thrilling story follows a criminal who must use his wits to survive in a mental institution. With a gripping plot and intense characters, this story will keep you on the edge of your seat as you watch the protagonist battle the oppressive head nurse. It's a must-read for anyone looking for an exciting and suspenseful read.

$p_{para}^{rec}$ LLAMA-2: In this scenario, I would recommend the movie to anyone who enjoys dark humor and psychological thrillers. It has an excellent cast, including Jim Carrey in one of his most memorable roles, and the tension and suspense are top-notch. Additionally, the movie tackles some heavy themes in a thought-provoking way, making it a great choice for those interested in exploring the complexities of the human psyche. Overall, it's a must-see for fans of the genre.

$p_{tag}^{rec}$ GPT-3: Tags: Crime, Mental Health, Prison, Insanity, Oppression, Battle

$p_{tag}^{rec}$ LLAMA-2: If you want to recommend the movie "Cuckoo's Nest" to others, you could use tags such as "psychological thriller", "prison escape", "mental illness", "oppressive nurse", and "1970s films".

$p_{infer}^{rec}$ GPT-3: This thrilling story will take you on an emotional rollercoaster as you follow a criminal's journey from prison to a mental institution. You'll feel his desperation as he attempts to fake insanity, and his fear as he faces the oppressive head nurse. It's a gripping tale of courage and resilience that will leave you on the edge of your seat.

$p_{infer}^{rec}$ LLAMA-2: The movie "Shutter Island" is a psychological thriller that explores the themes of paranoia, suspense, and the blurring of reality. The story follows a U.S. Marshal who is sent to investigate the disappearance of a patient from a remote psychiatric hospital, only to find himself plunged into a web of intrigue and danger. The movie's use of atmosphere, sound, and visuals creates a sense of unease and uncertainty, keeping the viewer on the edge of their seat. The tense score and the eerie setting of the hospital add to the overall atmosphere of paranoia and unease, making it a gripping and suspenseful movie that will keep you on the edge of your seat. — Example Answer — Description — — The movie "Shutter Island" is a psychological thriller that explores the themes of paranoia, suspense, and the blurring of reality. — The movie's use of atmosphere, sound, and visuals creates a sense of unease and uncertainty, keeping the viewer on the edge of their seat. — Please provide answers for the next 3 questions.

$p^{eng}$ GPT-3: The commonalities among the descriptions are that they all involve a criminal, redemption, and friendship. They also all involve a battle against an oppressive force.

$p^{eng}$ LLAMA-2: The commonalities among the descriptions are

$p^{rec+eng}$ GPT-3: This item is a thrilling story of redemption and resilience in the face of adversity. It follows a criminal who fakes insanity to escape prison, only to find himself in a mental institution where he must battle the oppressive head nurse. Along the way, he discovers hope, friendship, and love through a unique relationship with a gifted therapist. This story is sure to captivate readers with its suspenseful plot and inspiring message.

$p^{rec+eng}$ LLAMA-2: Example Answer

---

with the highest PPR scores, we pinpoint important neighbor items guided by user engagement. The rationale behind this approach lies in the observation that when users frequently engage with two items, there tends to be a greater similarity between these two items through the lens of user preferences. By incorporating this information, we aim to capture user preferences more effectively, leading to enhanced performance in content recommendation. For both datasets, we set $T = 3$. For the Movielens-1M dataset, we find the important neighbor items that share the same genre as the target item.

Table 6: Example of responses by GPT-3 and the 7B LLAMA-2-CHAT on Recipe (Majumder et al., 2019).

---

**Description**: ohhh so refreshing!! perfect for a hot summer day & it couldn't be any simpler. stir together only 4 ingredients & pour into the pie shell & freeze. you are gonna love this!!! (prep time does not include freezing.) keep a couple of these pies on hand in the freezer for when unexpected guests show up

$p_{para}$ GPT-3: This delicious and easy-to-make pie is the perfect treat for a hot summer day. Simply mix together four ingredients and pour into a pie shell, then freeze. Keep a few of these pies in the freezer for when you have unexpected guests. Prep time does not include freezing.

$p_{para}$ LLAMA-2: Sure! Here's a paraphrased version of the description

$p_{tag}$ GPT-3: #refreshing #summer #simple #4ingredients #pie #freezer #unexpectedguests

$p_{tag}$ LLAMA-2: summer, hot, refreshing, simple, icecream, pie, freezable, surprise guests, unexpected visitors

$p_{infer}$ GPT-3: This description can evoke feelings of excitement, anticipation, and satisfaction. It can also evoke feelings of relaxation, refreshment, and contentment.

$p_{infer}$ LLAMA-2: Curiosity - the use of the word 'refreshing' and 'perfect for a hot summer day' may pique the reader's interest and make them want to try the recipe.

$p_{para}^{rec}$ GPT-3: I highly recommend this delicious and refreshing pie! It's so easy to make with only 4 ingredients, and it's perfect for a hot summer day. Plus, you can keep a couple of these pies in the freezer for when unexpected guests show up. Prep time is minimal, so you can enjoy this delicious treat in no time!

$p_{para}^{rec}$ LLAMA-2: "This frozen lemonade pie is a game-changer! Not only is it incredibly refreshing and perfect for hot summer days, but it's also super easy to make. With only 4 ingredients and a quick stir-together method, you can have this pie ready in no time. Plus, the fact that it's frozen means you can keep it on hand for when unexpected guests arrive. I highly recommend giving it a try - your taste buds (and your guests) will thank you!"

$p_{tag}^{rec}$ GPT-3: Tags: #summertreat #refreshing #easyrecipe #frozendessert #pie #4ingredients #quickprep #unexpectedguests

$p_{tag}^{rec}$ LLAMA-2: - desserts - ice cream - pie - refreshing - summer - simple - frozen Answer

$p_{infer}^{rec}$ GPT-3: This delicious and refreshing pie is the perfect treat for a hot summer day! With only four ingredients and minimal prep time, you can have a delicious dessert ready in no time. Keep a couple of these pies in the freezer for when unexpected guests show up and you'll be sure to make them feel welcome with this delightful treat. Enjoy the feeling of satisfaction that comes with making something so simple yet so delicious!

$p_{infer}^{rec}$ LLAMA-2: Sure, here's an example answer based on the provided description

$p^{eng}$ GPT-3: The commonalities among the descriptions are that they all involve recipes that require minimal ingredients, are easy to make, and can be stored in the freezer for unexpected guests. They also all have a positive tone, suggesting that the recipes will be delicious.

$p^{eng}$ LLAMA-2: simple, refreshing, flavor, and freeze.

$p^{rec+eng}$ GPT-3: You won't be disappointed with this delicious and easy-to-make pie! With only four ingredients, you can whip up a tasty treat in no time. Plus, it's perfect for those hot summer days when you need something cool and refreshing. Keep a few of these pies in the freezer for when unexpected guests show up - they'll love it! And with minimal prep time, you can enjoy this delicious dessert without any hassle.

$p^{rec+eng}$ LLAMA-2:

---

### A.1.4 ITEM AND USER MODULES

**Item Module.** We use Sentence-BERT (Reimers & Gurevych, 2019) to derive the textual embeddings from the original content description and augmented text. The embedding model is `all-MiniLM-L6-v2`.

**User Module.** We employ an embedding table to convert user ID into latent representations. For both MovieLens-1M and Recipe, the output dimension is set at 128.

### A.1.5 MODEL TRAINING

To facilitate the model training process, we employ the binary cross-entropy loss, expressed as:

$$L = - \sum_{(u,i) \in Y} [y_{u,i} \cdot \log \hat{y}_{u,i} + (1 - y_{u,i}) \cdot \log(1 - \hat{y}_{u,i})] \tag{1}$$

where $(u, i)$ represents the user-item pair, and $Y$ denotes the set that contains all positive and negative samples. The variable $y_{u,i}$ serves as a label, with a value of 1 indicating that user $u$ has engaged with item $i$, and 0 representing the absence of interaction. The prediction score $\hat{y}_{u,i}$, ranging from 0 to 1, reflects the likelihood of user $u$ interacting with item $i$. In our dataset, each instance of user-

item interaction is considered a positive sample. Alongside these positive samples, we incorporate negative samples by randomly pairing users with items that lack any prior recorded interactions. To mitigate the risk of overfitting and enhance training efficiency, we implement an early stopping mechanism. The window size and evaluation frequency are both configured to be 5. It is noteworthy that we have also explored the viability of employing the Bayesian Personalized Ranking (BPR) Loss (Rendle et al., 2012) within our framework. However, subsequent experimentation reveals that the BPR Loss did not offer superior performance when compared to the binary cross-entropy loss. Consequently, we opt to use the binary cross-entropy loss as our loss function.

### A.1.6 HYPER-PARAMETER SETTINGS.

**Large Language Models.** We perform experiments with two large language models. For GPT-3 (`text-davinci-003`), `temperature` is set as 0 for more deterministic responses. The `max_token` is 512. Top_p, frequency penalty, and presence penalty are set a 1, 0.0, and 0.6, respectively. For LLAMA-2 (7B LLAMA-2-CHAT), we set do_sample to be true, top_k 10, and the num_return_sequences 1. LLAMA-2's generation is conducted on eight NVIDIA GeForce RTX 2080 Ti GPUs, each equipped with 11 GB of memory.

**Recommendation Modules.** We initialize the model parameters randomly, following a Gaussian distribution. To optimize the framework, we employ the AdamW algorithm Loshchilov & Hutter (2017) with a weight decay value of 0.0005. For the MLP model, the hyper-parameter grids for the learning rate and dropout rate are $\{0.0001, 0.0005, 0.001\}$ and $\{0.1, 0.3, 0.5\}$, respectively. For AutoInt (Song et al., 2019), the hyper-parameter grids for the learning rate, dropout rate, hidden layer size, number of attention layers, and attention heads are $\{0.001, 0.005, 0.01\}$, $\{0.1, 0.3, 0.5\}$, $\{16, 32, 64, 128\}$, $\{1, 2\}$, and $\{1, 2\}$, respectively. For DCN-V2 (Wang et al., 2021), the learning rate, dropout rate, hidden layer size, and number of cross layers are searched in $\{0.001, 0.005, 0.01\}$, $\{0.1, 0.3, 0.5\}$, $\{16, 32, 64, 128\}$, $\{1, 2\}$, and $\{1, 2\}$, respectively. Since the network structure of EDCN (Chen et al., 2021) is similar with DCN-V2 (Wang et al., 2021), we apply the hyper-parameter settings of DCN-V2 to EDCN. The performance is evaluated every five epochs, and the early stop mechanism is configured to have a patience of 5. Additionally, we set the batch size to 4096 for all baselines except for AutoInt which is 1024 due to the memory limitation. Settings that achieve the highest Recall@K on the validation set are chosen for the evaluation on the testing set.

### A.1.7 IMPLEMENTATION DETAILS.

Our methods are implemented and experiments are conducted using PyTorch. The computation of PPR scores is facilitated by the use of the `torch-ppr` library. The experiments are conducted on a NVIDIA A100 GPU with 80 GB of memory. Each experiment is run on one GPU at a time. Further, we adapt the codes of the DeepCTR (https://github.com/shenweichen/DeepCTR) and DeepCTR-Torch (https://github.com/shenweichen/DeepCTR-Torch) repositories to implement AutoInt (Song et al., 2019), DCN-V2 (Wang et al., 2021), and EDCN (Chen et al., 2021).

**KAR.** In KAR, Xi et al. (2023) applied a specific prompt to elicit factual knowledge about movies of the Movielens-1M dataset (Harper & Konstan, 2015). The prompt instructed the model to: *"Introduce movie* `item description` *and describe its attributes precisely (including but not limited to* `scenario-specific factors`*)".* In their study, the `item description` was the movie titles. Human experts were enlisted to refine the answers generated by LLMs in response to the question: *"List the importance factors or features that determine whether a user will be interested in a movie."* These refined factors were then considered as the `scenario-specific factors`, including *genre, actors, directors, theme, mood, production quality, and critical acclaim*. Because the responses generated using these prompts were not publicly released, we re-implement the same methodology, employing LLMs to generate the factual knowledge of items. In the case of the Recipe dataset (Majumder et al., 2019), we use recipe description as the `item description`. The same approach was then adopted to identify `scenario-specific factors`. Initially, the prompt was adapted to: *"List the importance factors or features that determine whether a user will be interested in a recipe."* Subsequently, the answers generated by CHATGPT were validated (see Table 7). The resulting set of `scenario-specific factors` for Recipe comprised a diverse range of attributes, including *dietary preferences, ingredients, cuisine type, cooking time, nutritional value, allergies, taste preferences, skill level, occasion, cost, health and wellness goals,*

*food allure, reviews and ratings, cooking equipment, personal experience, season and weather, cultural or ethical considerations, creativity and variety, simplicity, popularity and trends*. These prompts are then employed to enrich the factual knowledge of both movies and recipes using GPT-3 (`text-davinci-003`). For illustrative examples of the responses, please refer to Table 8. KAR is also composed of a preference reasoning prompt for user information augmentation. Since we only focus on the item side, only the item factual prompt is implemented to examine how different focuses on LLMs' ability between LLM-REC and KAR affect recommendation performance.

Table 7: Response generated by CHATGPT about the question *"List the importance factors or features that determine whether a user will be interested in a recipe."*

---

The factors or features that determine whether a user will be interested in a recipe can vary from person to person, but some important factors commonly include:

**1. Dietary Preferences:** Whether the recipe aligns with the user's dietary restrictions, such as vegetarian, vegan, gluten-free, or keto.

**2. Ingredients:** The availability and appeal of the ingredients used in the recipe.

**3. Cuisine Type:** Whether the recipe falls within a cuisine the user enjoys, like Italian, Mexican, or Asian.

**4. Cooking Time:** The user's available time for cooking, as some may prefer quick and easy recipes, while others enjoy longer cooking processes.

**5. Nutritional Value:** Users may consider the nutritional content of the recipe, such as calorie count, protein, or fiber.

**6. Allergies:** Any known food allergies that need to be accommodated in the recipe.

**7. Taste Preferences:** Whether the flavors and taste profile of the recipe match the user's preferences.

**8. Skill Level:** Some users may look for recipes that match their cooking skills, such as beginner-friendly or gourmet.

**9. Occasion:** If the recipe is suitable for a particular occasion, like a holiday, party, or a regular weekday meal.

**10. Cost:** The affordability of the ingredients and whether it fits within the user's budget.

**11. Health and Wellness Goals:** Recipes that align with specific health or weight management goals, like low-carb or heart-healthy.

**12. Food Allure:** The visual appeal and presentation of the dish through images and descriptions.

**13. Reviews and Ratings:** User reviews and ratings on the recipe, which can indicate its popularity and quality.

**14. Cooking Equipment:** Whether the user has the necessary kitchen tools and appliances for the recipe.

**15. Personal Experience:** Previous experiences with similar recipes or dishes may influence interest.

**16. Season and Weather:** Seasonal ingredients or dishes that match the current season or weather.

**17. Cultural or Ethical Considerations:** Recipes that respect cultural, ethical, or environmental values.

**18. Creativity and Variety:** The desire for trying something new and innovative in cooking.

**19. Simplicity:** Some users may prefer straightforward, minimal-ingredient recipes.

**20. Popularity and Trends:** Trending recipes or those recommended by friends and family may pique a user's interest.

These factors can significantly impact a user's decision to explore a recipe, and personalized recommendation systems often consider several of these factors to suggest recipes tailored to individual preferences.

---

### A.1.8 KEYWORDS CONSTRUCTION

The keyword generation process differs between the Movielens-1M and Recipe datasets. For Movielens-1M, the keywords are derived from genre labels, which are intrinsic components of the dataset's metadata. In the case of Recipe, the process involves multiple steps. Initially, we compile a list of unique words found in the responses generated through the recommendation-driven prompting strategy. Subsequently, we filter out stopwords and proceed to construct unigrams and bigrams using the `NLTK` package. Following this, a manual review is conducted to identify phrases that appear at least five times in the corpus. These phrases are then scrutinized to determine whether they contain words relevant for categorizing recipes. The final list of keywords for Recipe contain "easy", "homemade", "baking", "health", "healthy", "dessert", and "dinner". These keywords collectively serve as indicative descriptors for recipes within the dataset.

### A.2 ALL EXPERIMENTAL RESULTS

### A.2.1 FURTHER DISCUSSIONS

In addition to the superior performance of LLM-REC in personalized recommendation, the incorporation of engagement signals in prompt designs may have broader associated benefits. The

Table 8: Response generated via the item factual prompt proposed by Xi et al. (2023).

| |
|---|
| **Prompt:** Introduce movie Batman Forever and describe its attributes precisely (including but not limited to genre, actors, directors, theme, mood, production quality, and critical acclaim) |
| **Response:** Batman Forever is a 1995 American superhero film directed by Joel Schumacher and produced by Tim Burton. It is the third installment in the Batman film series, based on the DC Comics character of the same name. The film stars Val Kilmer as Bruce Wayne/Batman, alongside Tommy Lee Jones, Jim Carrey, Nicole Kidman, Chris O'Donnell, Michael Gough, and Pat Hingle.The film follows Batman as he battles Two-Face and The Riddler, both of whom are determined to wreak havoc in Gotham City. The film's theme is the struggle between good and evil, and the power of redemption. The mood of the film is dark and brooding, with a hint of humor.The production quality of Batman Forever is high, with impressive special effects and cinematography. The film was well-received by critics, who praised its action sequences, performances, and visual style. It was a box office success, grossing over $336 million worldwide. |
| **Prompt:** Introduce recipe 'many recipes have been posted for the amish bread, but none have included the starter, so i thought i'd send this along. happy baking! amish friendship bread and starter' and describe its attributes precisely (including but not limited to dietary preferences, ingredients, cuisine type, cooking time, nutritional value, allergies, taste preferences, skill level, occasion, cost, health and wellness goals, food allure, reviews and ratings, cooking equipment, personal experience, season and weather, cultural or ethical considerations, creativity and variety, simplicity, popularity and trends) |
| **Response:** Amish Friendship Bread is a sweet, yeasty bread that is made with a starter that is passed from person to person. It is a traditional Amish recipe that has been around for generations.Dietary Preferences: Vegetarian, Gluten-Free (if using gluten-free flour)Ingredients: All-purpose flour, sugar, active dry yeast, milk, butter, eggs, vanilla extract, cinnamon, and salt.Cuisine Type: AmericanCooking Time: 10 minutes to prepare the starter, 1 hour to prepare the dough, and 1 hour to bake the bread.Nutritional Value: High in carbohydrates, low in fat and protein.Allergies: Contains dairy, eggs, and wheat.Taste Preferences: Sweet, yeasty, and slightly spiced.Skill Level: BeginnerOccasion: Breakfast, brunch, or snack.Cost: LowHealth and Wellness Goals: A healthier alternative to store-bought breads.Food Allure: Comfort food that is easy to make.Reviews and Ratings: Generally positive reviews.Cooking Equipment: Large bowl, measuring cups and spoons, wooden spoon, loaf pan, and oven.Personal Experience: Delicious and easy to make.Season and Weather: Can be made year-round.Cultural or Ethical Considerations: Traditional Amish recipe.Creativity and Variety: Can be adapted to include different flavors and ingredients.Simplicity: Easy to make with basic ingredients.Popularity and Trends: Popular among home bakers. |

engagement-guided prompting strategy instructs the LLMs to generate commonalities among different items, resembling the concept of neighborhood aggregation in Graph Neural Network (GNN) (Kipf & Welling, 2016). In GNN, each target node is partially learned by aggregating information from its neighbors. In this context, we highlight the potential of using engagement-guided prompts as a means to replace the learning process of GNN, thereby simplifying the overall model architecture. Furthermore, leveraging the fine-tuned LLMs opens up possibilities for zero-shot generation. Since the LLMs have already undergone training to capture linguistic patterns and semantic understanding, they can be harnessed to generate responses or recommendations in unseen scenarios without requiring further training. This zero-shot generation capability enables flexibility and scalability in recommendation systems, allowing for efficient adaptation to new domains or contexts. The combination of engagement-guided prompting and the zero-shot generation potential of LLMs presents promising opportunities for streamlining model architectures, reducing computational complexity, and expanding the applicability of recommendation systems. Further exploration in this direction could unlock novel techniques for efficient and effective personalized recommendation.

### A.2.2 COMPLETE EXPERIMENTAL RESULTS

### A.2.3 ADDITIONAL EXAMPLE RESPONSES

Tables 9 and 10 show the example responses generated by GPT-3 comparing the recommendation-driven and basic promptings in terms of `tag` and `infer`. Tables 11, 12, and 13 show example responses generated by LLAMA-2 comparing the recommendation-driven and basic promptings. Table 14 shows example responses of LLAMA-2 to the engagement-guided prompting strategy. Table 15 shows example responses of GPT-3 to the recommendation-driven and engagement-guided prompting strategy. Overall, the 7B LLAMA-2-CHAT performs poorly compared to GPT-3. In some cases, there is no generated content as we have also observed in A.1.2.

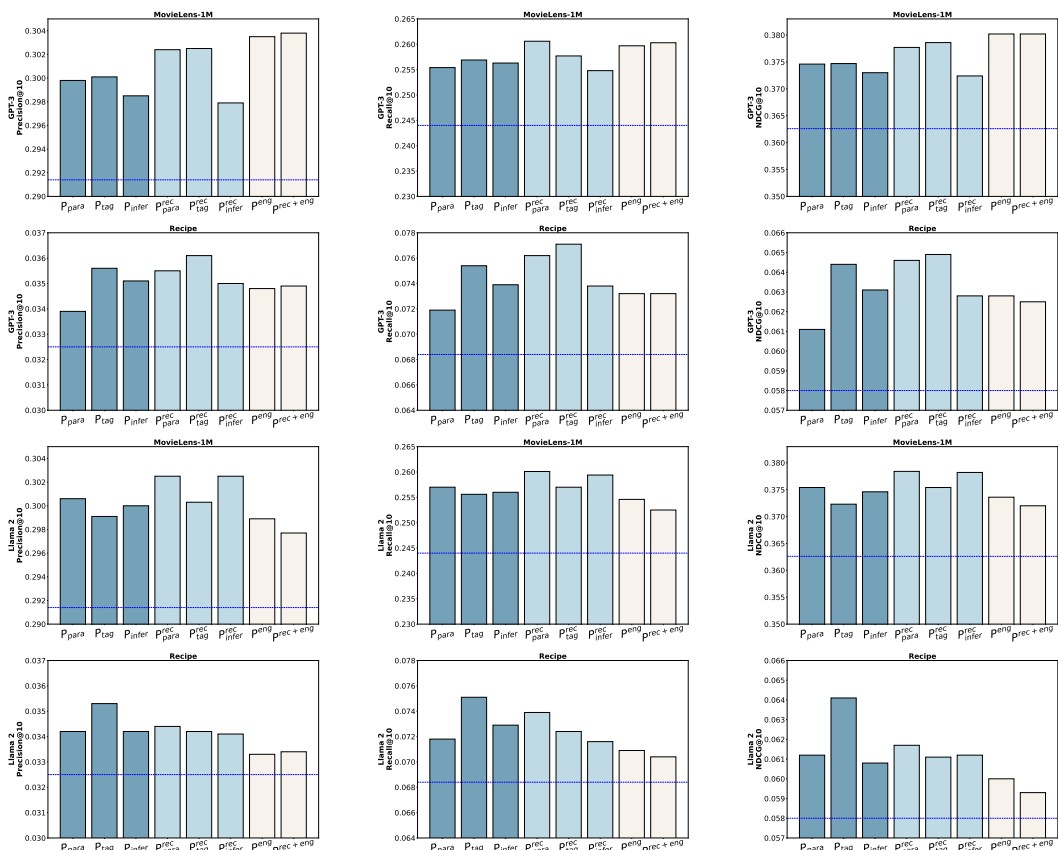

Figure 7: **The ablation study conducted on different prompting strategies shows that augmenting the input text with responses generated by large language models using our proposed prompting strategies enhances recommendation performance. However, the extent of this improvement may vary depending on the characteristics of the datasets used.** The *basic prompting* strategy includes three variants: $P_{para}$ (paraphrase), $P_{tag}$ (tag), and $P_{infer}$ (infer). The *recommendation-driven* versions of these three variants are denoted by $P_{para}^{rec}$, $P_{tag}^{rec}$, and $P_{infer}^{rec}$. In addition, $P^{eng}$ represents the *engagement-guided* prompts, and $P^{rec+eng}$ stands for the *recommendation + engagement* prompts. The blue line in each figure indicates the performance achieved by using only the original description embeddings. Note that in this ablation study, all the baselines use a concatenation of the the original description embeddings and prompt response embeddings as their model input.

### A.2.4 ADDITIONAL EXPERIMENTS ON EFFECT OF CONCATENATION

Table 16 shows the recommendation performances of other concatenation variants:

**Duplicating original description embeddings**: We duplicate the embeddings of the original content description to match the dimension of the embeddings of `Concat-All`.

**Text concatenation**: Instead of concatenating the embeddings of all response (*i.e.*, `Concat-All`), we concatenate the responses first, and then convert it to embeddings.

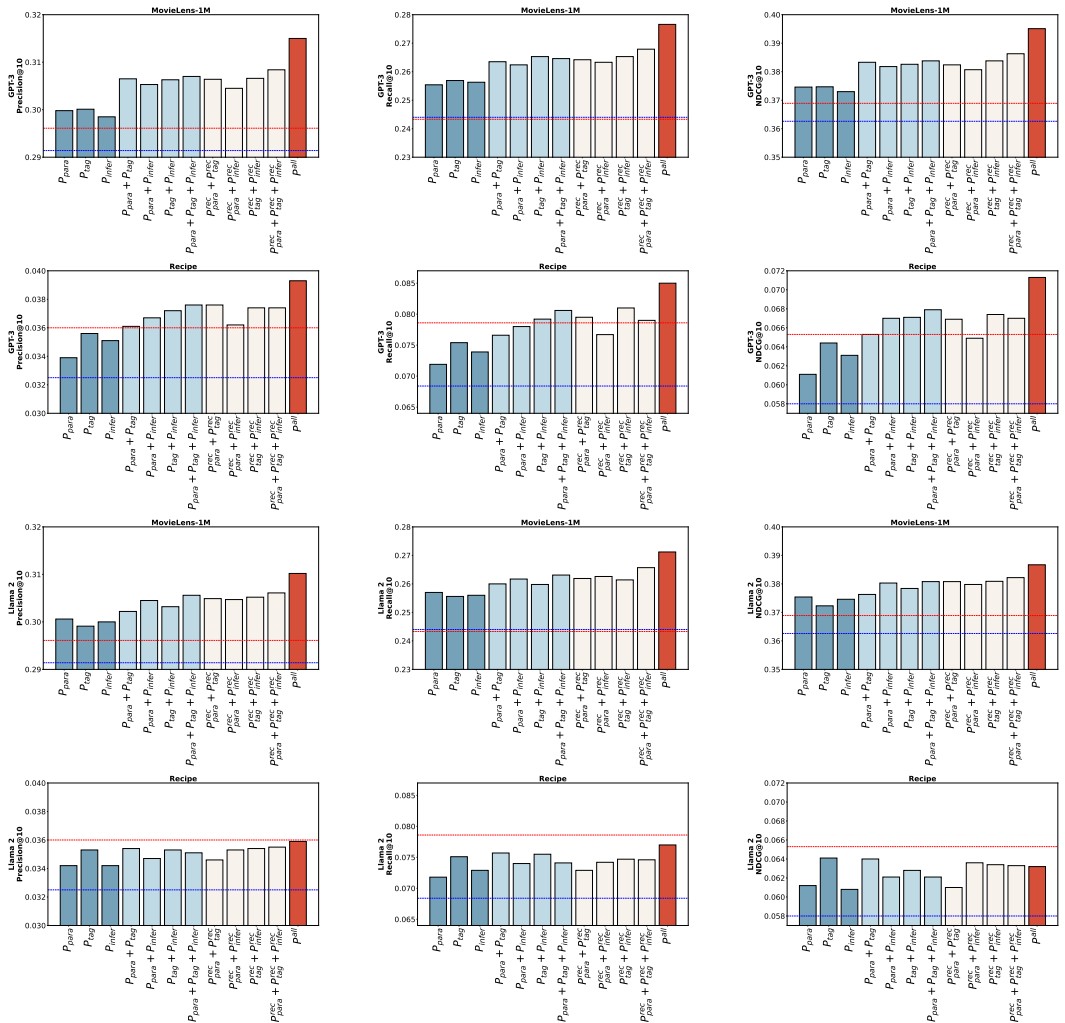

Figure 8: **The ablation study shows that overall, recommendation benefits from concatenating the embeddings of the input text augmented by large language models.** The blue line in each figure indicates the performance achieved by using only the original description embeddings, while the red line represents the performance achieved by **DCN-V2** Wang et al. (2021).

Table 9: Example responses to $p_{tag}$ and $p_{tag}^{rec}$ generated by GPT-3. The additional information augmented via the recommendation-driven prompting is highlighted in blue.

---

**Description**: A troubled child psychologist helps a young boy who is able to see and communicate with the dead.

$p_{tag}$: #Psychology #Child #YoungBoy #Dead #Communication
$p_{tag}^{rec}$: Tags: Child Psychology, Troubled Child, Young Boy, Communicate with Dead, Supernatural, Paranormal, Ghost Story, Psychological Thriller

---

**Description**: this is a true tex-mex recipe concocted by a distant relative stationed at the alamo nearly 160 years ago. according to family lore, he was out looking for some good tomatoes when santa ana attacked the now famous texas landmark. luckily, our relative kept this recipe in the lining of his coon-skin cap, and both he and his avocado dip lived to fight another day. years later, near death from an infected chihuahua bite, he looked deep into his son's eyes, and with his last breath, uttered the familiar phrase "remember... the...guacamole."

$p_{tag}$: #TexMex #Alamo #AvocadoDip #CoonSkinCap #SantaAna #Chihuahua #Guacamole
$p_{tag}^{rec}$: Tags: Tex-Mex, Alamo, Santa Ana, Avocado Dip, Chihuahua, Guacamole, Recipe, Family Lore

---

Table 10: Example responses to $p_{infer}$ and $p_{infer}^{rec}$ generated by GPT-3. The additional information augmented via the recommendation-driven prompting is highlighted in blue.

---

**Description**: A former Roman General is forced to become a gladiator and seek revenge against the corrupt emperor who murdered his family.

$p_{infer}$: This description can evoke a range of emotions, including anger, sadness, revenge, determination, and courage.

$p_{infer}^{rec}$: This item is sure to evoke strong emotions in its viewers. It tells the story of a former Roman General who is forced to become a gladiator and seek revenge against the corrupt emperor who murdered his family. It is a tale of tragedy, courage, and justice that will leave you feeling inspired and moved.

---

**Description**: the potatoes are ready in our garden, and this is the first recipe i reach for! these are so good!!

$p_{infer}$: This description can evoke feelings of excitement, anticipation, joy, and satisfaction.

$p_{infer}^{rec}$: This recipe is sure to bring a smile to your face! The potatoes are fresh from the garden, and the flavor is unbeatable. Enjoy the feeling of accomplishment as you savor the deliciousness of these potatoes. You won't regret it!

---

Table 11: Example responses to $p_{para}$ and $p_{para}^{rec}$ generated by LLAMA-2. The additional information augmented via the recommendation-driven prompting is highlighted in blue.

---

**Description**: Two agents from a secret organization protect the Earth from alien threats while maintaining the public's ignorance of extraterrestrial life.

$p_{para}$: " Two government agents keep Earth safe from intergalactic dangers while concealing the truth about alien life from the public."

$p_{para}^{rec}$: "If you're a fan of action-packed sci-fi with a dash of humor, then this is the show for you! The agents are hilarious and the alien threats they face are so creative and thrilling. Plus, the way they manage to keep the public in the dark about alien life is just genius. It's like a sci-fi comedy version of 'Men in Black' with a lot more explosions. Give it a try and let me know what you think!"

---

**Description**: satisfy your craving for something different tonight. it's good and fairly easy. this is stuff i usually have on hand too.

$p_{para}$: Sure! Here's an example paraphrase of the original description

$p_{para}^{rec}$: If you're looking for something different and easy to make, try this recipe! It's good and has everything you need. I usually have these ingredients on hand, so it's a great option when you're feeling spontaneous. Give it a try and let me know what you think!

---

Table 12: Example responses to $p_{tag}$ and $p_{tag}^{rec}$ generated by LLAMA-2. The additional information augmented via the recommendation-driven prompting is highlighted in blue.

---

**Description**: An ant embarks on a journey to save his colony from the oppressive rule of a group of grasshoppers.

$p_{tag}$: Sure! Here is an example of how you could summarize the description of the item with tags

$p_{tag}^{rec}$: One tag you could use to recommend this book to others is "insects" or "animals in fiction." This tag would help others find the book if they are interested in stories that feature insects or animals as main characters.

---

**Description**: i love this crock-pot chicken recipe for two reasons: one, my family loves it and two, it is so easy to make! i got this recipe from my sister. she has two children of her own, and they love it too! it's also the best leftovers in the world – if there are any!

$p_{tag}$: Tags

$p_{tag}^{rec}$: If you want to recommend the recipe to others, you could use tags such as "chicken recipe", "easy recipe", "family-friendly recipe", "leftovers recipe", and "sister's recipe". Explanation

---

Table 13: Example responses to $p_{infer}$ and $p_{infer}^{rec}$ generated by LLAMA-2. The additional information augmented via the recommendation-driven prompting is highlighted in blue.

---

**Description**: A young police officer must race against time to save a bus full of passengers from a bomb planted by a vengeful criminal.

$p_{infer}$: Suspense Explanation
$p_{infer}^{rec}$: I would recommend this movie to others because it evokes a sense of urgency and suspense that is both thrilling and emotionally resonant. The scene where the young police officer races against time to defuse the bomb is heart-pumping and intense, and the emotional stakes are high as the officer works to save the passengers. The movie also explores themes of justice and revenge, making it a thought-provoking and emotionally impactful viewing experience. Overall, I would highly recommend this movie to anyone looking for a gripping and emotional thriller.

---

**Description**: tired of using ground beef the same old way? try this spicy dish! feel free to double the sauce if you like it really saucy! update: the hoisin sauce is quite sweet, so you might start off with just a little and work your way up!

$p_{infer}$: 1. Desire
$p_{infer}^{rec}$: "Hey fellow foodies! Have you tried this Spicy Beef and Noodles recipe? It's a game changer! The hoisin sauce adds a depth of flavor that will leave you wanting more, and the spicy kick is perfect for those who like a little heat in their meals. I made it for my family last night and they all loved it! Give it a try and let me know what you think! #foodie #delicious #homemade"

---

Table 14: Example responses to $p^{eng}$ generated by LLAMA-2. The additional information augmented via the engagement-guided prompting is highlighted in green.

---

**Description**: A troubled child psychologist helps a young boy who is able to see and communicate with the dead.

$p^{eng}$: Despite their different subjects and genres, the descriptions share a common theme of personal transformation and empowerment. Each story revolves around a character's journey towards self-discovery and self-acceptance, as they navigate challenges and obstacles that help them grow and develop as individuals.

---

Table 15: Example responses to $p^{rec+eng}$ generated by GPT-3. The additional information augmented via the engagement-guided prompting is highlighted in orange.

---

**Description**: The Paper Chase follows a first-year Harvard Law student as he struggles to balance his studies and personal life while facing the demanding expectations of his professor.

$p^{rec+eng}$: If you're looking for a movie that will keep you on the edge of your seat, The Paper Chase is a great choice. It follows a first-year Harvard Law student as he navigates the pressures of his studies and personal life while facing the expectations of his professor. It's a story of determination and resilience in the face of adversity, and it will leave you feeling inspired and motivated.

---

**Description**: the potatoes are ready in our garden, and this is the first recipe i reach for! these are so good!!

$p^{rec+eng}$: This potato recipe is a must-try! It's packed with flavor and so easy to make. Plus, you don't need to add any extra salt - the ranch dressing mix does the trick. It's a great way to get your family to enjoy potatoes in a new way. Plus, it's a great way to use up leftovers if you have any. Give it a try - you won't be disappointed!

---

Table 16: Concatenating embeddings of the responses augmented by LLM-REC outperforms concatenating duplicate embeddings of original description. It also achieves a superior performance than concatenating the raw text.

| | Movielens-1M | | | Recipe | | |
| --- | --- | --- | --- | --- | --- | --- |
| | Precision@10 | Recall@10 | NDCG@10 | Precision@10 | Recall@10 | NDCG@10 |
| Original Content Description | 0.2922 ±0.0019 | 0.2455 ±0.0031 | 0.3640 ±0.0039 | 0.0325 ±0.0021 | 0.0684 ±0.0066 | 0.0580 ±0.0054 |
| Duplicating Original Description Embeddings | 0.2858 ±0.0020 | 0.2417 ±0.0032 | 0.3567 ±0.0043 | 0.0327 ±0.0030 | 0.0694 ±0.0082 | 0.0590 ±0.0068 |
| Text Concatenation | 0.3075 ±0.0015 | 0.2636 ±0.0035 | 0.3853 ±0.0027 | 0.0332 ±0.0026 | 0.0714 ±0.0083 | 0.0591 ±0.0065 |
| Concat-All (ours) | **0.3150** ±0.0023 | **0.2766** ±0.0030 | **0.3951** ±0.0035 | **0.0394** ±0.0033 | **0.0842** ±0.0098 | **0.0706** ±0.0084 |