# OpenReview forum: "LLM-Rec: Personalized Recommendation via Prompting Large Language Models"
_ICLR.cc/2024/Conference — ICLR 2024 Conference Withdrawn Submission_

### Official Review · Reviewer_u86v · 2023-10-30

**Soundness:** 3 good
**Presentation:** 4 excellent
**Contribution:** 3 good
**Rating:** 6
**Confidence:** 3

**Summary:**

This paper motivates that directly utilizing the item description as text input features is not yet optimal and proposes to leverage LLM to augment the text feature in order to improve the recommendation. More specifically, the paper proposes templates that prompt the LLM to derive a. recommendation oriented text feature, b.derive neighboring context aware textual summary, as well as the combination of an and b. To use the derived text feature, sbert model is used to encode the text into embeddings, which are used in the downstream recommendation module. Empirical experiments show by applying LLMs such as LLAMA2 and GPT(text-davinci-003) for item description augmentation, the basic MLP model out performs nontrivial baselines. A few other design choices are also empirically evaluated.


__Initial Recommendation__: Weak accept based on the reason that the proposed approach is mildly novel while effective on movie and recipe recommendation scenario. The motivation are sound and the proposed approach is justified in the paper.

**Strengths:**

* Although the idea of using LLM-augmented item description as textual feature for enhancing existing recommender models is in gist similar to the KAR approach, the proposed four prompt templates and the way of using the augmented textual feature is mildly novel in the context of LLM for recommendation.
* The design of prompt templates as well as the combination of different prompts are sound and justified.
* The empirical study supports the effectiveness claim of the proposed approach.

**Weaknesses:**

* The limitation discussion of the proposed approach seems to be missing from the paper. Inherited from LLMs, the proposed approach might have issues like computation efficiency, lacking awareness of latest event and the cost of keeping the model up-to-date, hallucination in general. Discussing these limitations would provide more practical insights about the proposed approach.
* The proposed approach applied on movie recommendation and recipe recommendation, explore its capability in e-commerce (Like in the KAR paper) or other scenario would further strengthen the generality claim of the paper.

**Questions:**

Please see the Weakness

---

> ### Author Response · Authors · 2023-11-23
> **Authors' response**
>
> We really appreciate the positive feedback from the reviewer. The suggestions about the limitations and generalizability are greatly appreciated which we will incorporate into our revision. They are helpful in improving our study.

---

### Official Review · Reviewer_vwfB · 2023-11-05

**Soundness:** 3 good
**Presentation:** 3 good
**Contribution:** 2 fair
**Rating:** 3
**Confidence:** 4

**Summary:**

This paper discusses approaches for augmenting prompting for LLMs in generating recommendations. Four prompting techniques were tested, basic prompting, recommendation-driven prompting, engagement-guided prompting, recommendation+engagement mixed prompting. Empirical results show that these augmentation techniques improve recommendation performance.

**Strengths:**

A nice part of the paper is the comprehensive evaluations of the 4 prompting techniques. There are some valuable insights obtained from the evaluations.

**Weaknesses:**

However, I do not see too much novelty from the paper beyond the evaluations. The paper specifically targets personalized recommendation evident from the title. I suppose the engagement-guided and recommendation+engagement mixed prompting can achieve some degree of personalization with the addition of important neighbors in the prompting. However, to achieve true personalization, identifying important neighbors is itself a critical problem. For example, what if there are many neighbors, how do you choose most important neighbors? Do you take raw engagement counts into account? Do you value more recent engagements? The paper does not have not many details on the topics at all. Without these technical details, the paper becomes just empirical evaluating of several prompting rules.

**Questions:**

How the proposed method solves personalized recommendation is unclear from the paper.

---

> ### Author Response · Authors · 2023-11-23
> **Authors' Response**
>
> We express our sincere gratitude to the reviewer for the valuable suggestions regarding the contribution of our study.
>
> We concur with the reviewer's perspective on the importance of gaining a deeper understanding of how LLM-Rec enhances personalization, particularly concerning significant neighbors. This insight is indeed crucial. However, it is also important to emphasize that our study's primary objective is to showcase an alternative method of utilizing LLMs. Our approach leverages the extensive knowledge and commonsense reasoning capabilities of LLMs to address existing challenges in personalization.
>
> As demonstrated in our experiments, LLMs have the potential to significantly improve recommendation performance. This improvement is primarily achieved by augmenting additional information to item descriptions that may initially be incomplete or insufficient. This augmentation aspect is a key differentiator in our approach, underlining the unique benefits of applying LLMs in the realm of personalized recommendations.

---

### Official Review · Reviewer_mt1T · 2023-11-06

**Soundness:** 2 fair
**Presentation:** 2 fair
**Contribution:** 2 fair
**Rating:** 3
**Confidence:** 4

**Summary:**

This paper examines the impact of prompting methods on large language models in the context of recommendation tasks, introducing four distinct techniques: basic prompting, recommendation-driven prompting, engagement-guided prompting, and recommendation+engagement prompting. The objective is to enhance textual representations with knowledge from pre-trained language models. However, the paper lacks substantial technical contributions. The provided prompt templates are simplistic and may not prove effective across different iterations of pre-trained language models due to their rapidly evolving nature. The experiments conducted are insufficient to substantiate the proposed methods' effectiveness, and the scope of application is limited to textual-rich datasets, excluding text-less datasets common in various scenarios.

**Strengths:**

1. Demonstrates the potential of large language models for data augmentation in Recommender Systems (RecSys).
2. Introduces four prompting methods to extract information from Large Language Models (LLMs).

**Weaknesses:**

1. Inadequate experimental validation of the proposed methods.
2. Limited applicability to datasets lacking textual information.
3. Fails to make a clear contribution to the field of RecSys.

**Questions:**

1. The main contribution revolves around prompt template design, which appears to be a manual fit for pre-trained LLMs. This approach lacks sustainability, considering the rapid changes in LLMs. Table 1 of the experiments highlights this issue, where even basic prompts outperform more intricate ones on Llama2. How can the authors address this limitation?

2. The application scope is restricted to textual-rich datasets. How does the proposed method fare in scenarios where datasets lack textual information, which is a common occurrence?

3. Why did the authors opt for partial ranking evaluation (1,000 items) instead of a full ranking approach, which might provide a more comprehensive assessment?

4. The comparison seems limited, especially considering the absence of a benchmark comparison with ID-based baselines such as DirectAU [1]. How does the proposed method fare against these established techniques?

[1] Wang, Chenyang, Yuanqing Yu, Weizhi Ma, Min Zhang, Chong Chen, Yiqun Liu, and Shaoping Ma. "Towards representation alignment and uniformity in collaborative filtering." In Proceedings of the 28th ACM SIGKDD Conference on Knowledge Discovery and Data Mining, pp. 1816-1825. 2022.

---

> ### Author Response · Authors · 2023-11-23
> **Authors' Response**
>
> We appreciate the valuable and constructive feedback from the reviewer. We are actively incorporating these suggestions into our revision.
>
> > The main contribution revolves around prompt template design, which appears to be a manual fit for pre-trained LLMs. This approach lacks sustainability, considering the rapid changes in LLMs. Table 1 of the experiments highlights this issue, where even basic prompts outperform more intricate ones on Llama2. How can the authors address this limitation?
>
> Thank you for pointing this out. We will add experiments of the prompts that have the same meaning but differ in wording to evaluate the consistency of LLM-Rec. The variant we use for Llama 2 is the Llama-2-Chat of 7B parameters. It is within our expectation that its generated text may not boost the performance of the recommendation performance as the generated text of GPT-3. However, the findings are still consistent. Adding extra information augmented by LLMs outperforms the model only takes the original item descriptions as input. Additionally, concatenating the augmented text can further improve the recommendation performance which allows single models such as MLP to achieve comparable and even better performance than other more complex feature-based models.
>
> >  The application scope is restricted to textual-rich datasets. How does the proposed method fare in scenarios where datasets lack textual information, which is a common occurrence?
>
> LLM-Rec aims to augment the input of the items that are relevant to rich textual content. The dataset itself does not necessarily have to contain rich textual information (The Movielens dataset does not contain movie description initially. We have shown how we can generate such description in the Appendix). The limitation lies in whether the item is associated with rich features, which we believe is another major issue in the recommendation area.
>
> > Why did the authors opt for partial ranking evaluation (1,000 items) instead of a full ranking approach, which might provide a more comprehensive assessment?
>
> Thank you for this suggestion. We follow previous studies for this evaluation method. We will add experiments of the full ranking approach in the revision.
>
> > The comparison seems limited, especially considering the absence of a benchmark comparison with ID-based baselines such as DirectAU [1]. How does the proposed method fare against these established techniques?
>
> Thank you for this suggestion. Our study aims to evaluate how LLMs can be used to improve the **content-based** recommendation which does not overlap with ID-based methods. Therefore, we did not compare our framework with the ID-based baselines.

---

### Official Review · Reviewer_UhyT · 2023-11-07

**Soundness:** 2 fair
**Presentation:** 2 fair
**Contribution:** 2 fair
**Rating:** 3
**Confidence:** 5

**Summary:**

This work explores the effects of different prompting strategies on text-based recommendation by enriching semantic information of textual item descriptions. Based on original item description, this work investigates four enhanced prompting strategies: (1) basic prompting, (2) recommendation-driven prompting, (3) engagement-guided prompting, and (4) recommendation-driven + engagement-guided prompting. Extensive experiments validate the effectiveness of generating additional textual information to improve performance in text-based recommendation task, which enables MLP to achieve comparable or even better performance than complex feature-based methods. Ablation studies also show that it’s possible to concatenate textual descriptions from more than one of prompting strategies to gain further improvement.

**Strengths:**

1.	LLM-Rec investigates diverse prompting strategies to enhance personalized text-based recommendations. The prompting strategies of recommendation-driven and engagement-guided exhibit the capability to tap into the language model's comprehension of both general and personalized item characteristics.
2.	Empirical experiments show discernible improvements in recommendation performance by incorporating augmented input text generated by LLMs.
3.	Provides insights into the potential of leveraging LLMs for personalized recommendation. Further experiments explore the potential of concatenating descriptions from multiple prompting strategies, enabling better performance.
4.	The overall presentation is clear.

**Weaknesses:**

1.	Some of the mistakes make it confused. (1) In Table 1, there is a wrong bold in the second column. (2) “Table 3”in the 7th page is typed as “Figure 3” by mistakes. (3) The citation at the end of the 5th page is not in the right format.
2.	The evaluation of the prompting strategies could benefit from increased interpretability. For instance, there is no experiment validating whether engagement-guided prompting effectively utilizes neighbor information to guide the LLM. It may seem counterintuitive to assert that important neighbor information is considered merely by including item names without a more detailed description and categorization.
3.	The paper lacks a detailed discussion on the potential limitations and challenges of using LLMs for personalized recommendation, such as computational complexity, model interpretability, and potential biases in the generated prompts.
4.	In the main experiment, Table 1 indicates that most of the time, combining both recommendation-driven and engagement-guided strategies leads to a decrease in performance, which appears to contradict the findings in Figure 6 and raises doubts about the generalization of these strategies.
5.	The contribution of this work can be further enhanced if it provides more explainability instead of focus more on improving performance with additional features.

**Questions:**

As a researcher, I aim to explore the practical situations where text-based recommendations can be applied, as this is crucial in determining the significance of this study. Could you kindly provide some practical situations?

**Details Of Ethics Concerns:**

No.

---

> ### Author Response · Authors · 2023-11-23
> **Authors' response (Part 1)**
>
> We thank the reviewer for the insightful and constructive feedback which are helpful in improving our study. We will respond to the highlighted weaknesses and questions in a point-by-point fashion.
>
> > Some of the mistakes make it confused. (1) In Table 1, there is a wrong bold in the second column. (2) “Table 3”in the 7th page is typed as “Figure 3” by mistakes. (3) The citation at the end of the 5th page is not in the right format.
>
> Thank you for catching these mistakes. We will fix them in the revision.
>
> > The evaluation of the prompting strategies could benefit from increased interpretability. For instance, there is no experiment validating whether engagement-guided prompting effectively utilizes neighbor information to guide the LLM. It may seem counterintuitive to assert that important neighbor information is considered merely by including item names without a more detailed description and categorization.
>
> We believe there is a misunderstanding about the engagement-guided prompting. In fact, the engagement-guided prompting strategy does NOT focus on item names, it includes the descriptions of other important neighbors. In Figure 1 and Section 2 (the last sentence of the 4th paragraph), we show that the prompt of the engagement-guided prompting is “Summarize the commonalities among the following descriptions: ‘description’; ‘descriptions of other important neighbors’”. Here the ‘description’ refers to the description of the target item, and ‘descriptions of other important neighbors’ refers to the item descriptions of other important neighbors. It does not represent item names.
>
> > The paper lacks a detailed discussion on the potential limitations and challenges of using LLMs for personalized recommendation, such as computational complexity, model interpretability, and potential biases in the generated prompts.
>
> We appreciate this suggestion and will include a detailed discussion on the potential limitations and challenges of using LLMs for personalized recommendation in Appendix. Here we briefly discuss the potential limitations and challenges:
>
> **Computational Complexity:** The LLM-Rec framework focuses on enhancing input for personalized recommendations through generating item descriptions. This approach addresses the problem of original item descriptions often being incomplete or insufficient for effective recommendations. The primary computational load in LLM-Rec arises during the augmentation phase including the output text length, number of items, and the concatenation of text embeddings.
>
> For output text length, our findings indicate that selecting important words for inclusion, rather than incorporating all response words, can lead to improved recommendation performance, as evidenced in Table 3. Future research could explore the balance between the number of words generated and the resulting performance enhancements.
>
> For the number of items, to mitigate this, we propose potential strategies to reduce the need for augmentation across numerous descriptions. One approach is developing a model that evaluates whether an original item description already possesses sufficient information for recommendations. This avenue warrants further exploration in subsequent studies to decrease computational demands.
>
> For the concatenation of text embeddings, as demonstrated in Table 2 and Figure 6, combining the embeddings of augmented text substantially enhances performance. However, this also increases the input dimension for the recommendation module, potentially raising computational costs. A solution could be to opt for a simpler model structure, such as an MLP, which we have implemented in LLM-Rec, to manage this increase in computational complexity.
>
> **Model Interpretability:** LLM-Rec introduces an innovative approach to model interpretability by enabling direct comparisons between variations in generated prompts and corresponding shifts in recommendation performance. This relationship is illustrated in our research, as showcased in Figures 4 and 5, as well as Tables 3, and 9-16. Nonetheless, we propose that delving into a more granular, word-level analysis could yield deeper insights. By examining both the generated responses and the subsequent recommendation module, we can gain a more nuanced understanding of how specific textual changes influence overall model behavior, enhancing our comprehension of model interpretability.
>
> **Potential Biases in generated responses:** The presence of potential biases in the prompts generated by LLMs is a significant concern, primarily because these biases are deeply ingrained in the data used to train the LLMs. This challenge calls for a multifaceted approach, including algorithmic adjustments, ethical guidelines, and continuous monitoring for bias, which are important future directions.

---

> > ### Author Response · Authors · 2023-11-23
> > **Authors' response (Part 2)**
> >
> > > In the main experiment, Table 1 indicates that most of the time, combining both recommendation-driven and engagement-guided strategies leads to a decrease in performance, which appears to contradict the findings in Figure 6 and raises doubts about the generalization of these strategies.
> >
> > Thank you for pointing out this seemingly inconsistent performance issue. In fact, the findings do not contradict each other. The recommendation-driven + engagement-guided prompting strategy instructs LLMs to generate descriptions with two objectives (i.e., recommendation-driven, engagement consideration). It leaves to LLMs to determine the amount of information of each component. However, Figure 6 shows the experimental results of directly concatenating the embeddings of each separately generated prompts and leaves to the following recommendation module to determine the amount of the information of each component.
> >
> > > The contribution of this work can be further enhanced if it provides more explainability instead of focus more on improving performance with additional features.
> >
> > Thank you for this suggestion. To provide insights into explainability, we directly compare the generated text of different prompting strategies and the corresponding recommendation performances (Figures 4,5, and Tables 9-16). The differences in the generated text are highlighted. To deepen our understanding of how different generated words influence recommendation performance. We further design an experiment comparing p_mask and p_keyword (Table 3). Again, we appreciate your suggestion and will include more case studies in the revision.
> >
> >
> > > As a researcher, I aim to explore the practical situations where text-based recommendations can be applied, as this is crucial in determining the significance of this study. Could you kindly provide some practical situations?
> >
> > Thank you for this interesting and important question. There are a few practical situations that can benefit from LLM-Rec. As we discussed in Introduction, the items whose description contains incomplete or insufficient information can benefit from the augmentation of LLMs. There is more improvement in the recommendation performance when the content is not well-defined or categorized. Further, we believe the text-based recommendations can be applied more effectively in scenarios where the recommendation relies on the textual content analysis. These textual content can be item descriptions, user reviews, and content summary *etc*.